behaviour/evolution

communication, sociality, social behaviour, dominance style, vocal

**Authors for correspondence:**
Eithne Kavanagh
e-mail: eithne.kavanagh@ntu.ac.uk
Katie Slocombe
e-mail: katie.slocombe@york.ac.uk

# Dominance style is a key predictor of vocal use and evolution across nonhuman primates

Eithne Kavanagh[1,2], Sally E. Street[3], Felix O. Angwela[5], Thore J. Bergman[6], Maryjka B. Blaszczyk[7], Laura M. Bolt[8], Margarita Briseño-Jaramillo[9,10], Michelle Brown[11], Chloe Chen-Kraus[12], Zanna Clay[4], Camille Coye[13,14], Melissa Emery Thompson[15], Alejandro Estrada[16], Claudia Fichtel[17,19], Barbara Fruth[20,21,22], Marco Gamba[23], Cristina Giacoma[23], Kirsty E. Graham[1,24], Samantha Green[25,26], Cyril C. Grueter[25,26,27], Shreejata Gupta[1], Morgan L. Gustison[28], Lindsey Hagberg[29], Daniela Hedwig[30], Katharine M. Jack[31], Peter M. Kappeler[17,32], Gillian King-Bailey[31], Barbora Kuběnová[33], Alban Lemasson[14], David MacGregor Inglis[34], Zarin Machanda[35], Andrew MacIntosh[33], Bonaventura Majolo[36], Sophie Marshall[1], Stephanie Mercier[37,38], Jérôme Micheletta[39,40], Martin Muller[15], Hugh Notman[41], Karim Ouattara[42], Julia Ostner[18,19,43], Mary S. M. Pavelka[44], Louise R. Peckre[17,19], Megan Petersdorf[45], Fredy Quintero[37], Gabriel Ramos-Fernández[46,47],

Martha M. Robbins[48], Roberta Salmi[49], Isaac Schamberg[29], Valérie A. M. Schoof[31,50], Oliver Schülke[18,19,43], Stuart Semple[34], Joan B. Silk[51], J. Roberto Sosa-Lopéz[52], Valeria Torti[23], Daria Valente[23], Raffaella Ventura[53], Erica van de Waal[38,54], Anna H. Weyher[55], Claudia Wilke[1], Richard Wrangham[29], Christopher Young[56,57,58], Anna Zanoli[23], Klaus Zuberbühler[35,59], Adriano R. Lameira[59,60] and Katie Slocombe[1]

[1]Department of Psychology, University of York, Heslington, York YO10 5DD, UK

[2]Department of Psychology, Nottingham Trent University, Chaucer Building, 50 Shakespeare St, Nottingham NG1 4FQ, UK

[3]Department of Anthropology and [4]Department of Psychology, Durham University, South Road, Durham DH1 3LE, UK

[5]School of Agricultural and Environmental Sciences, Mountains of the Moon University, PO Box 837, Fort Portal, Uganda

[6]Departments of Psychology, EEB, University of Michigan, Ann Arbor, MI 48109, USA

[7]Department of Anthropology, University of Texas at Austin, 2201 Speedway Stop C3200, Austin, TX 78712, USA

[8]Department of Anthropology, University of Waterloo, 200 University Avenue West, Waterloo, Ontario, Canada N2 L 3G1

[9]Instituto de Biologia, Universidad Nacional Autonoma de México (UNAM), Circuito exterior s/n, Ciudad Universitaria, Copilco, Coyoacán, Mexico City 04510, Mexico

[10]Centro Interdisciplinario de Investigación para el Desarrollo Integral Regional Unidad Oaxaca (CIIDIR), Instituto Politécnico Nacional, Hornos No. 1003, Col. Noche Buena, Municipio de Santa Cruz Xoxocotlán, Oaxaca 71230, Mexico

[11]Department of Anthropology, University of California, 552 University Road, Santa Barbara, CA 93106-3210, USA

[12]Department of Anthropology, Yale University, 10 Sachem Street, New Haven, CT 06511, USA

[13]College of Life and Environmental Sciences, University of Exeter, Penryn Campus Treliever Road, Penryn TR10 9FE, UK

[14]Human and Animal Ethology (EthoS), University of Rennes, Normandie University, CNRS, EthoS - UMR6552, Campus de Beaulieu, 263 Avenue du Général Leclerc, 35000 Rennes, France

[15]Department of Anthropology, University of New Mexico, 500 University Boulevard NE, Albuquerque, NM 87131, USA

[16]Field Research Station Los Tuxtlas, Institute of Biology, National Autonomous University of Mexico, Circuito interior s/n, Ciudad universitaria, Delegacion coyoacan, Mexico City CP 04510, Mexico

[17]Behavioral Ecology and Sociobiology Unit, German Primate Center and [18]Research Group Primate Social Evolution, German Primate Center, Leibniz Institute for Primate Research, Kellnerweg 4, 37077 Göttingen, Germany

[19]Leibniz ScienceCampus Primate Cognition, Kellnerweg 4, 37077 Göttingen, Germany

[20]School of Biological and Environmental Science, Liverpool John Moores University, Liverpool L3 3AF, UK

[21]Centre for Research and Conservation, Royal Zoological Society of Antwerp, 2018 Antwerp, Belgium

[22]Department of the Ecology of Animal Societies, Max Planck Institute of Animal Behavior, Bücklestraße 5, 78467 Konstanz, Germany

[23]Department of Life Sciences and Systems Biology, University of Turin, via Accademia Albertina, 13, 10123 Turin, Italy

[24]School of Psychology & Neuroscience, University of St Andrews, St Andrews, KY16 9JP, UK

[25]School of Human Sciences, [26]UWA Africa Research and Engagement Centre and [27]Centre for Evolutionary Biology, School of Biological Sciences, The University of Western Australia, 35 Stirling Highway, 6009 Crawley, Western Australia, Australia

[28]Department of Integrative Biology, University of Texas at Austin, 2415 Speedway, Austin, TX 78712, USA

[29]Department of Human Evolutionary Biology, Harvard University, 11 Divinity Avenue, Cambridge, MA 02138, USA

[30]K. Lisa Yang Center for Conservation Bioacoustics, Cornell Lab of Ornithology, Cornell University, 159 Sapsucker Woods Road, Ithaca, NY 14850, USA

[31]Department of Anthropology, Tulane University, 6823 St. Charles Avenue, New Orleans, LA 70118, USA

[32]Department Sociobiology/Anthropology, Johann-Friedrich-Blumenbach Institute of Zoology and Anthropology, University Göttingen, Kellnerweg 6, 37077 Göttingen, Germany

[33]Primate Research Institute, Kyoto University, 41-2 Kanrin, Inuyama, Aichi 484-8506, Japan

[34]Department of Life Sciences, University of Roehampton, Holybourne Avenue, London SW15 4JD, UK

[35]Department of Anthropology, Tufts University, 5 The Green, Medford, MA 02155, USA

[36]School of Psychology, University of Lincoln, Lincoln, Brayford Wharf East LN5 7TS, UK

[37]Department of Comparative Cognition, Institute of Biology, University of Neuchâtel, Rue Emile-Argand 11, 2000 Neuchâtel, Switzerland

[38]Inkawu Vervet Project, Mawana Game Reserve, Swart Mfolozi 3115, South Africa

[39]Department of Psychology, Centre for Evolutionary and Comparative Psychology, University of Portsmouth, King Henry Building, King Henry I Street, PO1 2DY Portsmouth, UK

[40]Macaca Nigra Project, Tangkoko Reserve, PO Box 1495, Bitung, Indonesia

[41]Anthropology, Faculty of Humanities and Social Sciences, Athabasca University, Athabasca, Canada

[42]Centre Suisse de Recherches Scientifiques en Côte d'Ivoire, 01 BP 1303 Abidjan 01, Ivory Coast

[43]Department of Behavioral Ecology, Johann-Friedrich-Blumenbach Institute for Zoology and Anthropology, University Goettingen, Göttingen, Germany

[44]Department of Anthropology and Archaeology, University of Calgary, 2500 University Drive NW, Calgary, Alberta, Canada T2N 1N4

[45]Department of Anthropology, New York University, 25 Waverly Place, New York, NY, USA

[46]Instituto de Investigaciones en Matemáticas Aplicadas y en Sistemas, Universidad Nacional Autónoma de México, Circuto Escolar 3000, C.U., 04510 Mexico City, Mexico

[47]UPIITA, Instituto Politécnico Nacional, Avenida Instituto Politécnico Nacional 2580, La Laguna Ticoman, 07340 Mexico City, Mexico
[48]Department of Primatology, Max Planck Institute for Evolutionary Anthropology, Deutscher Platz 6, 04103 Leipzig, Germany
[49]Department of Anthropology, University of Georgia, 355 S. Jackson Street, Athens, GA 30602, USA
[50]Department of Biology, York University, Keele Campus, 4700, Keele Street, Toronto, ON Canada, M3J 1P3
[51]School of Human Evolution and Social Change, Arizona State University, Tempe, AZ 85287, USA
[52]CONACYT-Centro Interdisciplinario de Investigación para el Desarrollo Integral Regional Unidad Oaxaca (CIIDIR), Instituto Politécnico Nacional, Hornos No. 1003, Col. Noche Buena, Santa Cruz Xoxocotlán, Oaxaca 71230, Mexico
[53]Scottish Primate Research Group, Division of Psychology, School of Social and Health Sciences, University of Abertay Dundee, Dundee, Scotland
[54]Department of Ecology and Evolution, University of Lausanne, 1015 Lausanne, Switzerland
[55]Department of Anthropology, University of Massachusetts Amherst, 240 Hicks Way #217, Amherst, MA 01003, USA
[56]Endocrine Research Laboratory, Mammal Research Institute, Faculty of Natural and Agricultural Science, University of Pretoria, Hatfield, Pretoria 0028, Republic of South Africa
[57]Applied Behavioural Ecology and Ecosystems Research Unit, University of South Africa, Pretoria, Florida 1710, Republic of South Africa
[58]Department of Psychology, University of Lethbridge, Alberta, Canada T1K6T5
[59]School of Psychology and Neuroscience, University of St. Andrews, South Street, St. Mary's Quad, South Street, St. Andrews KY16 9JP, UK
[60]Department of Psychology, University of Warwick, University Road, Humanities Building, Coventry CV4 7AL, UK

EK, 0000-0001-7202-005X; SES, 0000-0001-8939-8016; TJB, 0000-0002-9615-5001; MBB, 0000-0002-9157-3052; LMB, 0000-0002-8275-6543; MB, 0000-0002-2995-1745; CC-K, 0000-0002-4136-077X; ZC, 0000-0002-3016-1732; MET, 0000-0003-2451-6397; AE, 0000-0002-6107-9109; CF, 0000-0002-8346-2168; BF, 0000-0001-9217-3053; MG, 0000-0001-9545-2242; KEG, 0000-0002-7422-7676; CCG, 0000-0001-8770-8148; MLG, 0000-0002-1162-8966; KMJ, 0000-0003-3569-8544; PMK, 0000-0002-4801-487X; AL, 0000-0001-8418-5601; ZM, 0000-0001-7060-7949; BM, 0000-0002-0235-3040; SM, 0000-0002-4257-1456; JM, 0000-0002-4480-6781; JO, 0000-0001-6871-9976; LRP, 0000-0002-0065-8529; GR-F, 0000-0001-7175-3905; OS, 0000-0003-0028-9425; SS, 0000-0003-0452-8104; JRS-L, 0000-0002-0120-0704; VT, 0000-0002-6908-1203; DV, 0000-0001-6086-5135; CY, 0000-0001-8919-2093; AZ, 0000-0002-9402-5617; ARL, 0000-0003-3071-4824; KS, 0000-0002-7310-1887

Animal communication has long been thought to be subject to pressures and constraints associated with social relationships. However, our understanding of how the nature and quality of social relationships relates to the use and evolution of communication is limited by a lack of directly comparable methods across multiple levels of analysis. Here, we analysed observational data from 111 wild groups belonging to 26 non-human primate species, to test how vocal communication relates to dominance style (the strictness with which a dominance hierarchy is enforced, ranging from 'despotic' to 'tolerant'). At the individual-level, we found that dominant individuals who were more tolerant vocalized at a higher rate than their despotic counterparts. This indicates that tolerance within a relationship may place pressure on the dominant partner to communicate more during social interactions. At the species-level, however, despotic species exhibited a larger repertoire of hierarchy-related vocalizations than their tolerant counterparts. Findings suggest primate signals are used and evolve in tandem with the nature of interactions that characterize individuals' social relationships.

# 1. Introduction

In the quest to understand the evolution of human and non-human animal behaviour, a 'multiple levels of analysis' approach has often been advocated [1]. Such an approach requires analyses to span different levels of a system (e.g. individual, family, group, species), however, in practice, most studies focus on a single level of analysis (e.g. variation between individuals or across species). A number of major theories purport that the evolution of communication and language can be understood in terms of the constraints and pressures of the social environment [2–4]. Empirical studies that examine patterns of association across multiple levels of analysis in tandem are likely to be vital in generating a comprehensive picture of the evolutionary landscape of sociality and communication, yet such studies are rare. Similarly, while social complexity has been proposed to be a key driver of variation in communication across animal taxa [4], empirical tests of this hypothesis have relied heavily on both species-level analysis and using group size as a measure of social complexity, neglecting other important aspects of sociality [5]. Group size cannot explain individual variation in communication within a group, as it does not capture the nature and type of interactions happening within groups which can result in varying constraints and pressures on individuals [6]. To offer new insights, we conducted a multiple-levels-of-analysis study with a focus on a measure that is related to the nature rather than the number of social relationships: dominance style.

The dominance style of social relationships is a potentially important, yet rather neglected, factor for understanding communication. Many animal species who live in social groups have asymmetrical

dominance relationships, and for these species, there is variation in their dominance 'style'; i.e. the degree to which behaviour between dominant and subordinate partners is consistent with the direction of the dominance asymmetry between them [7]. We use the terms 'despotic' and 'tolerant' to refer to relationships in which the dominant partner asserts their dominance to a greater or lesser degree, respectively, meaning only dominant partners can 'give' tolerance (e.g. by allowing or being unable to prevent aggression from the subordinate), while subordinate partners can only 'receive' tolerance (note that these definitions are distinct from the despotic/egalitarian continuum in socioecological theories, e.g. [8]). Although dominance style is considered a relationship-level construct [9], empirical studies indicate that there is considerable variation in dominance style between certain primate species (e.g. macaque species [10]) and some variation has been documented between groups of a single species [11]. It is important, however, not to overlook the variation in dominance style that may occur at the level of the individual within a species (e.g. variation in the degree to which (i) each dominant individual in a group is tolerant of subordinates and (ii) each subordinate individual receives tolerance from dominants). We, therefore, sought to test whether dominance style affected communication at both the species- and individual-level.

At the individual-level, we predicted that tolerant relationships might be associated with a higher rate of communication for both dominant and subordinate individuals. Within despotic relationships, dominant individuals may be more likely to use the threat of force to achieve goals, or to be automatically allocated resources or receive services without need for communication [12]. Within tolerant relationships, there may be more pressure on dominant individuals to communicate to make requests, as their status alone is less likely to be effective in achieving goals, or to affiliate with subordinates, as this is likely to play a role alongside aggression in gaining access to food and mating opportunities, and maintaining rank [13,14]. For subordinate individuals, we suggest that the cost of communicating may be reduced in tolerance compared to despotic relationships. If signals are misunderstood or perceived as a transgression of the hierarchy by dominant partners, they could be more likely to be punished with physical aggression in despotic relationships. Giving and receiving tolerance may result in greater usage of communication, but for very different reasons. It is, therefore, crucial that these two perspectives of dominance style are examined separately, and this study is the first empirical attempt to do this.

At the species-level, we predicted that greater tolerance could be associated with greater diversity in species' communication systems (e.g. larger repertoire sizes) as this is predicted by Freeberg *et al.*'s [4] social complexity hypothesis. Globally, the greater social complexity of tolerant species should result in more complex communication systems, often operationalized as the number of discrete signals in a species' repertoire [4], compared to despotic species. Freeberg *et al.* [4] argued that while despotism severely limits the extent of possible relationships within a group, tolerance involves more reversals of interactions, and this greater diversity of directional connections between individuals indicates greater complexity from a social network perspective. The interactions within tolerant relationships also likely require more complex social information processing due to greater social uncertainty [15]. The proposed function of asymmetrical dominance relationships is to help to minimize injury associated with conflicts over resources, by making the outcomes of interactions more predictable [16]. Thus, a clear prediction of Freeberg *et al.*'s [4] hypothesis is that species comprising relatively tolerant relationships should have more complex communication systems than those comprising relatively despotic relationships, due to greater uncertainty in the outcome of their interactions.

In order to disentangle the individual- and species-level processes that underpin any relationships between dominance style and communication, it is necessary to directly quantify dominance style in a rich comparative behavioural dataset at multiple levels of analysis across a wide range of species. Understanding communication in non-human primates (henceforth 'primates') is particularly important for informing theories of language evolution due to their close phylogenetic relationship to humans [3,4], thus previous studies and ours have focussed on the primate order. A few existing studies have attempted to examine how dominance style relates to communication use and repertoires, finding mixed results [17–25]. Tolerance within the macaque genus may be related to larger facial and gestural repertoires [18,22,23]. By contrast, small-scale studies suggest dominance style may have a negative or no relationship with vocal repertoire size [17,21,24]. There is, however, some indication that greater tolerance is linked to lower vocal usage [17], to greater structural complexity [25] and differential patterns of acoustic similarity between dyads in macaques [19], and to the patterns and contexts of the use of specific vocal types in baboons [20,24]. However, these studies included only between two and eleven primate species, generally within a single genus, and typically only considered a single level of analysis. Further, in many of these studies it remains unclear which social variable is most closely linked to communication as dominance style was confounded by other social dimensions (e.g. in [17] the most despotic of the three species considered was the only species with a multi-male, multi-female social

structure, with the most tolerant species consisting of single male-multi-female groups), and dominance style was not always directly measured alongside communication in the same individuals.

In the current study, we aimed to advance our understanding of how dominance style relates to vocal usage and evolution. We examined dominance style and vocal communication at multiple levels of analysis in a broad sample of wild groups of primate species, by analysing a large observational dataset on aggression, grooming, feeding and vocal behaviour. Vocal rate was calculated from focal observations of individuals across a range of contexts. While dominance style may vary between sexes, we were interested in obtaining a holistic characterization of the nature of social interactions that individuals have with all group members, rather than with a subset of those members (e.g. individuals of the same sex). Dominance style has not been investigated in many primate species, so we sought to use a broad set of measures that have been successfully used in previous primate research. We thus measured dominance style using three behavioural measures focussed on agonistic interactions that have been previously used extensively in macaques (aggression symmetry, counteraggression and aggression intensity; [7,26–28]). These measures were combined into a single composite dominance style score at the species-level. We also conducted an additional analysis using feeding proximity as a potential indicator of tolerance, as this is a frequently used measure in research with various other primate and non-primate species (e.g. chimpanzees, bonobos, lemurs, wolves, dogs, birds; [29–33]). It is currently unclear whether this 'tendency to be in close proximity around valued resources' directly aligns with the definition of tolerance in macaque research, although tolerant macaques may also feed in closer proximity than despotic macaques [34]. In any case, it is a good potential candidate for predicting vocal rate for many of the same reasons outlined above.

At the individual-level, we examined separately the tolerance given by dominants (given tolerance) and the tolerance received by subordinates (received tolerance) as there are different reasons for expecting dominant and subordinate individuals within tolerant relationships to communicate more frequently than those in despotic relationships. We predicted that both given and received tolerance, and feeding proximity, would be associated with a higher rate of vocalizing across contexts.

At the species-level, we predicted that a dominance style index (based on aggression symmetry, counteraggression and aggression intensity) would be associated with two aspects of vocal repertoires. Repertoire size is one indication of communicative complexity [4,26], so we predicted that more tolerant species (indicated by a higher dominance style index score) would have larger vocal repertoires. We also predicted that more despotic species (indicated by a lower dominance style index score) would have more vocal signals in their repertoires associated with the establishment and/or maintenance of the hierarchy (i.e. dominance/appeasement signals). Despotic species should have a greater need to manage and reinforce the hierarchy, putting greater selection pressure on signals associated with the hierarchy.

# 2. Methods

## 2.1. Subjects

Our sample included 111 wild, habituated, non-provisioned groups of 26 primate species, all of which have some degree of dominance asymmetry in social relationships documented (see electronic supplementary material, S1 for details of species and groups in sample). As aggression data were required for calculating three out of our four measures, individuals and species with very low aggression rates are not well represented in our analysed samples, after applying minimum criteria for inclusion (see electronic supplementary material, S2).

## 2.2. Dominance style data

### 2.2.1. Data collection

As this study was a collaborative effort, it includes a mix of data that were obtained specifically for the current study, and data already collected for unrelated projects. For this reason, not all behavioural measures were available for all groups, and there was variation in data collection methods and behavioural definitions (e.g. different inter-bout intervals). We dealt with these issues by constraining our measures to those unaffected by varying definitions (e.g. we did not include aggression rate, as this would have been affected by variation in bout definition) and by excluding datasets without the

**Table 1.** Description of four dominance style variables.

| dominance style variables | description |
| --- | --- |
| aggression symmetry | the degree to which the direction of aggressive bouts within a dyad tends to be symmetrical, as opposed to one individual initiating aggression with the other more often than vice versa. Measured by the directional inconsistency index (DII) of aggression; the proportion of bouts within a dyad in which the roles of aggressor and victim occurred in the least frequent direction |
| counteraggression | the percentage of aggressive bouts in which the victim retaliates against the initial aggressor |
| aggression intensity | the percentage of aggressive bouts in which the aggressor uses physical contact |
| feeding proximity | the percentage of scans in which an individual was within 1 m of another independent individual while feeding[a] |

[a]Feeding proximity was excluded from any models or calculations of given and received tolerance and was treated separately as an individual-level dominance style measure potentially encompassing both given and received tolerance. This is because for many datasets we could not ascertain whether individuals other than the nearest neighbour were within 1 m, so could not reliably determine the percentage of scans within 1 m of a higher or lower ranking individual. Feeding proximity was not calculated at the species-level as there were too few species with data for this measure to merit doing this.

minimum amount of data required for each analysis. Electronic supplementary material, S3 indicates the data available for each group/species, and reasons for missing data.

Aggression behaviour was recorded using focal, all-occurrence or ad libitum sampling methods (see electronic supplementary material, table S4 for aggression ethograms, and S5 for observational methods and bout definitions used for each group). Feeding proximity was recorded using instantaneous point samples of the distance between a focal individual and its nearest neighbour while the focal individual was in a feeding context (see electronic supplementary material, table S5 for intervals of scans across groups).

### 2.2.2. Dominance style measures

From the data, we extracted four dominance style variables which are described in table 1. Aggression symmetry (i.e. aggression directional inconsistency index; DII), counteraggression, and aggression intensity, are all commonly used dominance style variables in macaques [7,27,28,35] while feeding proximity is frequently used with other species [29–33].

At the individual-level, (i) 'given tolerance' (the tendency of an individual to assert their dominance over subordinate partners) and (ii) 'received tolerance' (the degree to which the individual's dominant partners tend to assert their dominance over them) versions of aggression symmetry, counteraggression and aggression intensity were extracted. A single individual-level version of (iii) feeding proximity was extracted as it may be conceptually different to the other measures, and because the identity of all nearby individuals was not always known, which would be required to calculate given and received tolerance for this measure. Additionally, (iv) species-level versions of aggression symmetry, counteraggression and aggression intensity were calculated.

The given tolerance measures were calculated by including only behavioural interactions with lower-ranking partners, while the received tolerance measures were calculated by including only behavioural interactions with higher-ranking partners. For species-level analyses, we created a dominance style composite measure that captured variation across the three measures of aggression DII, counteraggression and aggression intensity in a single value (a similar approach to composite indices of association or friendship previously calculated in primates [36,37]). To create this composite measure the following steps were taken: (i) we calculated the mean value for each measure of all individuals in a group, then (ii) the mean value of all groups in the species. (iii) Z-scores of the three measures were then calculated and (iv) finally the mean of the three z-scores for each species was calculated. More details of how measures were calculated are provided in electronic supplementary material, S2.

To determine the dominance ranks of individuals in each group we calculated modified David's scores [38] using SOCPROG on Matlab [39] based on decided aggressive bouts (winner determined

by the victim fleeing or submitting). We necessarily had to exclude individuals without the required flee/submission data (resulting in the exclusion of one species).

We provide details on an additional, exploratory dominance style measure (groom symmetry) and its relationship with vocal measures in electronic supplementary material, S2 and S10. Groom symmetry (i.e. groom DII) is a previously unused but potentially useful measure of dominance style. High-ranking primates tend to receive more grooming [40,41] so greater symmetry in the direction of grooming within a dyad could indicate a more tolerant relationship.

## 2.3. Vocal communication data

### 2.3.1. Vocal rate

For each group, vocal rate data were collected during an observation period that overlapped with that of the dominance style data collection. From focal observations, we measured the number of vocal bouts produced by each individual per hour. In line with many previous studies of primate vocalizations, we defined a vocal bout as all vocalizations produced by the individual within 30 s of one another [42–45]. A minimum of 2 h focal observation time was required for the inclusion of an individual. Note that vocal rate is not considered a measure of communicative complexity.

### 2.3.2. Vocal repertoires

In the absence of an extensive acoustic dataset being available for each of our study groups, in line with previous studies [26,46], we extracted data on the vocal repertoire of each species from previous literature. We followed McComb & Semple's [26] criteria for selection of repertoires and call types as closely as possible (see electronic supplementary material, table S6A for publications used per species and necessary deviations from their criteria for our sample of species). This ensured the highest degree of comparability across species that is possible from the current repertoire literature. One species (*Propithecus verreauxi*) was excluded for all repertoire analyses as it had no published repertoire.

We extracted two vocal repertoire measures for each species: the overall repertoire size, which we consider a measure of communicative complexity, and the number of hierarchy-related calls in the repertoire (a subset of the overall repertoire size). Calls described as occurring in an appeasement or dominance context were classed as hierarchy-related signals (see electronic supplementary material, table S7 for definitions of these contexts).

## 2.4. Data analysis

### 2.4.1. Individual-level analyses

To test our individual-level predictions that (i) given tolerance and (ii) received tolerance would predict vocal rate, we constructed two models with vocal rate as the dependent variable, either the three given or the three received tolerance measures (aggression symmetry, counteraggression and aggression intensity) and group size as predictor variables, with group and species as random effects (nested). We fitted these models in R in a Bayesian framework, with the package MCMCglmm [47] using a Gaussian distribution with a log10 transformation on the dependent variable. We also tested whether (iii) feeding proximity predicted vocal rate. As this variable could potentially encompass both given and received tolerance, it was entered into a separate GLMM as a fixed effect. For all models, we used default diffuse normal prior probability distributions for the predictor variables and commonly used inverse Wishart priors for the random effects and residual variance (setting $V = 1$ and $v = 0.002$; [47]). See table 2 for a summary of these models, and see electronic supplementary material, S8 for species and groups included in all models. In general, for relatively small (approx. 30 or fewer) samples of species, the phylogenetic signal cannot be reliably estimated [48] but our inclusion of species as a random effect does, however, control for non-independence of observations within species (as in Sol *et al.* [49] for example). All model diagnostics showed reasonable fit; see electronic supplementary material, S9 for further details on all models including diagnostic plots.

### 2.4.2. Species-level analyses

To test whether dominance style predicts (iv) the overall vocal repertoire size, or (v) the number of hierarchy-related vocalizations in a species' repertoire, we fitted two species-level models with one of

**Table 2.** Summary of the main individual- and species-level models. MCMC, Monte Carlo Markov chain; GLMM, generalized linear mixed model; PGLS, phylogenetic generalized least squares.

| main individual-level models (Bayesian MCMC GLMMs) | | | |
|---|---|---|---|
| model | dependent variable | random effects | fixed effects |
| (i) given tolerance full model[a] (N = 181 individuals from 16 species) | vocal rate | –group<br>–species | –aggression DII<br>–counteraggression<br>–aggression intensity<br>–group size |
| (ii) received tolerance full model[a] (N = 178 individuals from 16 species) | vocal rate | –group<br>–species | –aggression DII<br>–counteraggression<br>–aggression intensity<br>–group size |
| (iii) feeding proximity model (N = 240 individuals from 8 species) | vocal rate | –group<br>–species | –feeding proximity<br>–group size |
| main species-level models (frequentist PGLS models with Pagel's $\lambda = 1$) | | | |
| | dependent variable | fixed effects | |
| (iv) overall repertoire size (N = 21 species) | N calls in repertoire | –dominance style composite index<br>–group size | |
| (v) hierarchy-related call repertoire size (N = 21 species) | N hierarchy-related calls | –dominance style composite index<br>–group size | |

[a]Given tolerance form of dominance style variables used as fixed effects in the given tolerance model, and the received tolerance form in the received tolerance model.

the two repertoire size measures as the response variables, and group size (mean of all groups in species) and the dominance style composite index as predictors. Here, we used frequentist PGLS models in the caper R package [50,51], as we had no need for individual and group-level effects, and it allowed us to fix Pagel's $\lambda$ to 1 avoiding issues of statistical power associated with estimating phylogenetic signal in small samples of species [48]. This highly conservative approach assumes that phylogeny has the maximum possible influence over the species' residual variance in the model. It was not possible to follow the same approach for the individual-level models as the MCMCglmm R package does not allow for phylogenetic signal (estimated using the heritability parameter, $h^2$) to be fixed to the maximum possible value. One species (*Papio kindae*) present in our species-level dataset was not included in the 10 k trees phylogeny [52]. To avoid having to exclude this species from our PGLS analyses, we grafted it onto the 10 k trees phylogeny as a sister lineage to *Papio cynocephalus* with a divergence time of 1.99 Myr, based on estimates provided by the Time Tree of Life resource [53]. See table 2 for summary of main models.

# 3. Results

## 3.1. Individual-level analyses

Description of the variation within species in terms of vocal rate and dominance style measures can be found in electronic supplementary material, table S6B.

### 3.1.1. Given tolerance

In support of our prediction that dominant individuals who are more tolerant of subordinate partners vocalize more frequently than their despotic counterparts, we found that individuals with higher

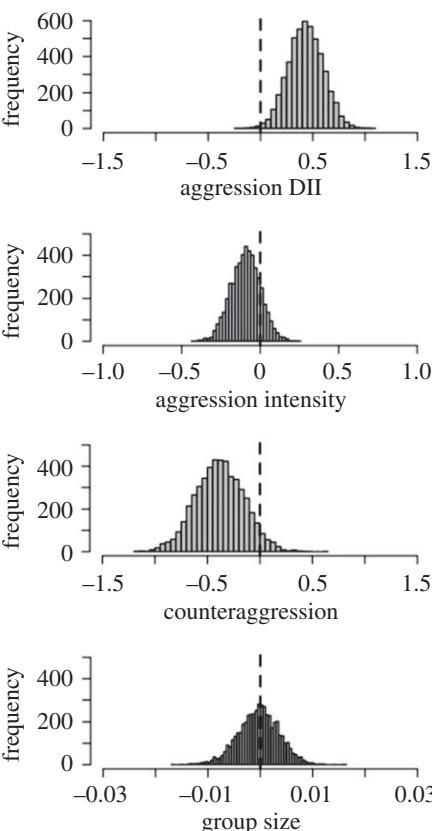

**Figure 1.** Model estimates from Bayesian analyses showing a positive effect of aggression symmetry on the vocal rate. Histograms show posterior distributions of $\beta$ coefficients for the effects of aggression symmetry, counteraggression, aggression intensity (given tolerance measures) and group size on the vocal rate. The distribution for aggression symmetry is shifted substantially away from zero, indicating evidence for an effect in the corresponding direction. Distributions for the other variables are centred closer to zero, indicating little or no evidence for effects.

**Table 3.** Results of MCMC model with given tolerance measures as fixed effects ($N$ individuals = 181, $N$ groups = 43, $N$ species = 16). Random effects of group and species explained 35% and 42% of variance, respectively, while fixed effects explained 6% of variance. Note that as species and group explained a relatively large amount of variance (likely due to phylogenetic ancestry and ecological conditions), by necessity the behavioural measures must explain a relatively small amount, but this does not mean they are biologically insignificant.

|  | variable | posterior mean $\beta$ | 1–95% CI | u-95% CI | pMCMC |
|---|---|---|---|---|---|
| fixed effects | aggression symmetry | 0.42 | 0.09 | 0.79 | 0.02 |
|  | counteraggression | −0.22 | −0.75 | 0.34 | 0.44 |
|  | aggression intensity | −0.09 | −0.28 | 0.12 | 0.43 |
|  | group size | <0.001 | −0.01 | 0.01 | 0.84 |

given tolerance, in terms of more symmetrical aggression, vocalized more frequently (see table 3 for full GLMM results). However, no clear relationship was found between vocal rate and the other two given tolerance measures or group size. Figure 1 displays the posterior distributions for the $\beta$ coefficients for all fixed effects in this model.

We ran an additional model with aggression symmetry as the only fixed effect and vocal rate as the dependent variable (with groups and species as random effects) on a larger sample of 253 individuals from 19 species. This model also indicated that aggression symmetry was an important predictor of vocal rate (see electronic supplementary material, table S10C), indicating that this finding is generalizable across our full sample of individuals with aggression symmetry data. Overall, our results indicate that dominant individuals whose aggression is more symmetrical with that of lower-ranking partners vocalize at a higher rate.

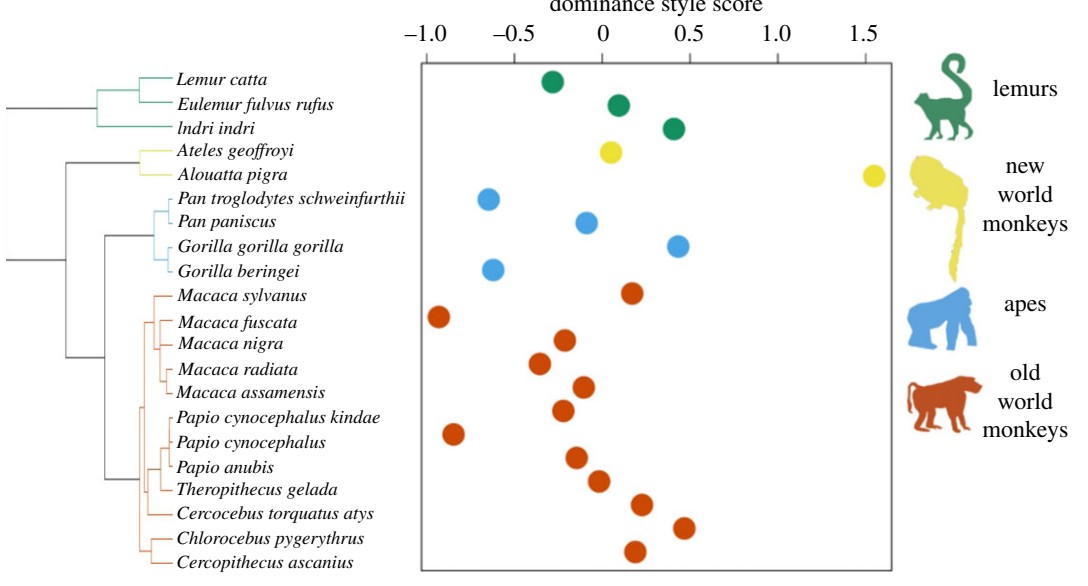

**Figure 2.** Dominance style composite index values across species in the sample for which phylogenetic information was available. Positive index scores indicate tolerance, while negative index scores indicate despotism. The cladogram on the left shows the phylogeny from 10 k trees.

**Table 4.** Results of MCMC model with received tolerance measures as fixed effects (N individuals = 178, N groups = 32, N species = 16). Random effects of group and species explained 25% and 51% of variance, respectively, while fixed effects explained 4% of variance.

|  | variable | posterior mean $\beta$ | 1–95% CI | u-95% CI | pMCMC |
|---|---|---|---|---|---|
| fixed effects | aggression symmetry | 0.14 | −0.20 | 0.48 | 0.42 |
|  | counteraggression | −0.05 | −0.67 | 0.53 | 0.88 |
|  | aggression intensity | −0.02 | −0.22 | 0.17 | 0.81 |
|  | group size | 0.004 | −0.004 | 0.01 | 0.33 |

### 3.1.2. Received tolerance

Contrary to our prediction that subordinate individuals would be free to vocalize more frequently if dominant partners are more tolerant of them, our GLMMs indicated no clear relationship between vocal rate and any of our received tolerance measures (table 4). Hence, we found no strong evidence that subordinate individuals vocalized more frequently if dominant partners were more tolerant towards them.

### 3.1.3. Feeding proximity

Contrary to our prediction that individuals who tend to stay close to other group members during feeding would vocalize more frequently, our GLMM indicated that feeding proximity was not associated with vocal rate (N individuals = 232, N groups = 37, N species = 8, Posterior mean $\beta = -$ 0.004, pMCMC = 0.97). Random effects of group and species explained 5 and 75% of the variance, respectively. This means that we found no support for our prediction that individuals who frequently feed close to other individuals vocalize more frequently.

### 3.2. Species-level analyses

Our behavioural dataset allowed us to calculate a dominance style index for 21 of the species in our sample as they met the minimum requirements for inclusion; figure 2 illustrates the spread of values of the dominance composite index across species in the primate phylogenetic tree.

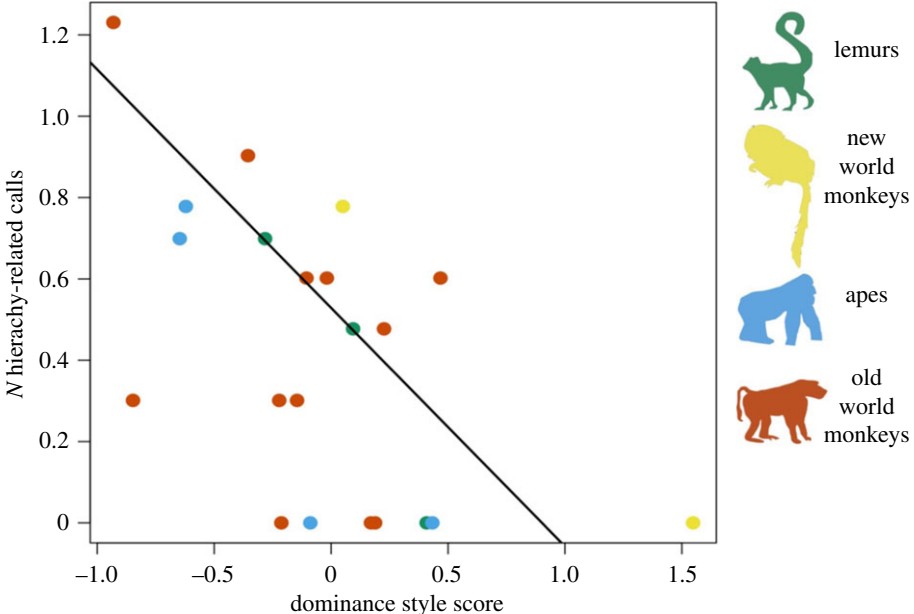

**Figure 3.** The relationship between the number of hierarchy-related calls in the species repertoire with their dominance style composite index score. Positive index scores indicate tolerance, while negative index scores indicate despotism. Each point represents a species. Number of hierarchy-related calls is log10 transformed in line with the PGLS models. The black line of fit is from the PGLS model (v) which assumes maximum phylogenetic signal.

We found no support for our prediction (iv) that more tolerant species would have larger vocal repertoires ($N = 21$, $\beta = -0.01$, $p = 0.891$), but we did find support for our prediction (v) that more despotic species had more hierarchy-related calls ($N = 21$, $\beta = -0.59$, $p = 0.001$). Figure 3 displays the relationship between the number of hierarchy-related calls and the dominance style composite index.

To see which individual dominance style measure most strongly predicts the number of hierarchy-related signals, we constructed separate models including each of the three dominance style measures and group size separately as predictor variables, instead of the dominance style composite index. We found that a larger repertoire of hierarchy-related calls was predicted by lower counteraggression ($N = 16$ species, $\beta = -2.36$, $p = 0.01$), and higher aggression intensity ($N = 17$ species, $\beta = 1.22$, $p < 0.001$), both of which indicate greater despotism, while aggression DII ($N = 21$ species, $\beta = 0.14$, $p = 0.940$) and group size ($N = 25$ species, $\beta < 0.001$, $p = 0.952$) did not significantly predict repertoire size of hierarchy-related signals. Overall, our results indicate that more despotic species have richer repertoires of hierarchy-related calls, but contrary to our expectations, our results provide no evidence that more tolerant species have more complex communication systems in the form of larger vocal repertoires overall.

## 4. Discussion

Using a rich behavioural dataset from 26 primate species, we quantified dominance style for the first time in many of these species and found evidence that dominance style was related to vocal communication at both individual and species levels. Notably, we found that more despotic species had richer repertoires of hierarchy-related calls in their evolved vocal systems, but that individuals who were more tolerant of lower-ranking partners vocalized at a higher rate. Overall, our findings suggest that the strictness of the dominance relationships of individuals and species provides important context for understanding primate vocal usage and evolution.

We found different effects of dominance style on vocal usage for dominant and subordinate partners, highlighting the importance of examining both sides of dyadic relationships separately. Our results indicated that individuals vocalized at a higher rate if they had more symmetrical aggression with subordinate partners (an indication of greater tolerance). This is consistent with our suggestion that tolerant individuals likely have a greater need to communicate in order to manage interactions with more uncertain outcomes, to affiliate with a wider range of individuals, or to request resources and services, rather than relying on higher dominance status and/or brute force to obtain them. However, we did not find that subordinate individuals vocalized more if their relationships with dominant

partners were more tolerant. For subordinate individuals, there could be a similar need to negotiate and affiliate regardless of the tolerance of dominant partners. Compared to dominant individuals, subordinates are likely to be under increased pressure in any social environment to take the perspective of others, a skill that requires complex social information processing [54]. As such, receiving tolerance may not increase social complexity for individuals as strongly as giving tolerance, hence no clear effect of received tolerance on the rate of communicating. By examining both components of dominance style relationships, we found that tolerance is likely to be linked to vocal communication more strongly as a result of increased pressure than the alleviation of constraints.

Our study is the first to provide evidence that more despotic species have richer repertoires of dominance and appeasement vocalizations. This result relates to Preuschoft & van Schaik's [16] finding that despotism predicted the existence of formal facial signals of dominance status in a species, as well as findings related to the much-studied 'silent bared-teeth' facial display. This signal is used symmetrically and flexibly in tolerant species but asymmetrically and in narrow contexts in despotic species [55]. Similarly, it is possible that the hierarchy-related vocalizations in our despotic species could have homologues in tolerant species, with despotism related to the asymmetry in their use between dominant and subordinate individuals, rather than driving the evolution of the signals themselves. Our findings open up exciting avenues for future research testing this possibility.

Our species-level findings did not provide support for the social complexity hypothesis [4]. Contrary to our predictions, we did not find that tolerant species had larger overall vocal repertoires nor did we find that group size predicted any vocal measure. A recent study by Rebout et al. [25] found that in agonistic and affiliative contexts, but not neutral contexts, the diversity of call types produced by two tolerant macaque species was greater than two despotic species. This may indicate that our failure to find an effect on tolerance on overall repertoire size is because we were not able to take the context of emission into account. It is also possible that when considering a broad range of primate species, dominance style is not a sufficiently stable social variable across generations and evolutionary time to have an effect on evolved communication systems. Currently, the stability of dominance style across generations is largely unknown and could vary across species. Game theory models indicate that a despotic strategy may be more stable than a tolerant strategy [56], which could explain why we found stronger evidence for despotism being related to evolved communication systems than tolerance in our broad sample of species.

Measurement differences could also explain why in contrast to Rebout et al. [25] we did not find that tolerant species had larger vocal repertoires and in contrast to Semple & McComb [26], we found no support for group size predicting vocal repertoire size. While we empirically measured dominance style, Rebout et al. [25] relied on previously made characterizations of macaque species tolerance levels based on only female behaviour. This is a limitation that persists in the majority of research testing how dominance style relates to other variables, despite evidence of intra-species variation in dominance style [11]. Indeed, while crested macaques have been characterized as highly tolerant in the literature and in Rebout et al. [25], they did not emerge as particularly tolerant based on our behavioural assessment (figure 2). Conversely, while Rebout et al. [25] were able to employ acoustic clustering methods to quantify vocal diversity, this optimal approach was simply not feasible to implement with our large sample of species, so we used published vocal repertoires instead. A concerted research effort is required to ensure that comparable objective vocal repertoires based on the acoustic measurement and statistical classification of call types are available for each species [57]. While we used a similar approach to measuring vocal repertoire as McComb & Semple [34], some group size estimates differ considerably in species that feature in both samples (see [58] for review of group size variability), which could explain why we did not replicate their findings. For instance, while they report a group size of 125 for bonobos, there were just 24 in the group in our sample. Our study suggests it may be worth revisiting whether primate species who live in larger groups do have larger vocal repertoires. These measurement discrepancies between studies highlight the importance of employing comparable methods which capture variation within as well as between species.

We interpret our results with the caveat that more validation of dominance style measures across primate species is needed to confirm the validity of our findings. Ours is the first attempt to measure dominance style on a continuous scale across the primate order, so it is perhaps not surprising that we did not find that all dominance style measures predicted communication in our sample. While we largely based our measures on those successfully used within the macaque genus, the measures that best capture the construct of dominance style are likely to vary due to species differences in managing dominance relationships, and future research needs to investigate the optimum set of measures for each species. As the majority of our dominance style measures could only be applied to species that engaged in regular within-group aggression, our sample excluded species who exhibited very low

levels of aggression. A different approach is, therefore, needed to understand how dominance style relationships are managed in species with very low aggression rates. When considering primate species who do engage in regular within-group aggression, aggression is a key tool in establishing and managing dominance relationships within a hierarchy. As aggression asymmetry is a defining feature of dominance and a despotic dominance style, measures relating to aggression symmetry may be the most robust indication of dominance style in primates.

While a great collaborative effort was required to generate the dataset for this study, it could be further improved with additional data and variables. As a result of missing data, not all individuals or species were included in all models, although data were available for the majority of species in our main findings (e.g. 19 of the 26 species were included in our model with aggression symmetry and vocal rate). As with all unimodal approaches, our singular focus on vocal communication meant we did not account for communication produced through other communicative modalities, although some species or individuals may rely more heavily on facial, gestural or olfactory communication than vocal signals. A more holistic, multimodal approach to characterizing communication in future would, therefore, be beneficial [59,60]. Additionally, while we included group size as a control variable in our models, the number of social and ecological factors that may relate to vocal communication is vast, and it was not feasible to account for all of these. This was due in no small part to the fact that many ecological variables, such as habitat visibility and food abundance can be challenging to quantify in directly comparable ways [61]. We view our study as an important first step that can be built on by future research. We hope that our findings encourage further examination into the strictness of dominance relationships and communication across the animal kingdom. Asymmetrical dominance relations are observable not only in primates, but in a wide range of animal taxa (e.g. birds [62], elephants [63], pigs [64]). As such, how communication is related to dominance style in these species should be investigated. Although our results are consistent with our causal hypothesis, our correlational design cannot establish the direction of causality and we suggest future simulations of experimental studies do so. Future research could also examine whether sex differences in dominance style within a species explain sex differences in vocal use and repertoires. Additionally, the pattern of results we identify in primates raises the question of how dominance style may relate to human language and communication. Humans are thought to have lived in highly tolerant societies in our recent evolutionary past [65], so how dominance style might have contributed to the evolution of human language is a promising prospect for further examination.

In this study, we found that dominance style predicts both individual- and species-level variation in vocal communication. This is the first to assess this relationship across multiple primate genera with a multiple-levels-of-analysis approach, and our findings provide valuable insights into the pressures shaping vocal communication. Our behavioural dataset will support future examination of other potential correlates of dominance style, such as cognitive or ecological factors, or to further explore its relationship with communication by using a multimodal approach [66]. If we are to understand the effects of social behaviour on the evolution of communication, we need to investigate this from multiple angles across diverse taxa, and to move beyond a narrow focus on group size by incorporating measures of the nature of social interactions, such as dominance style.

Ethics. This study complied with the ASAB/ABS guidelines for the use of animals in research and was granted ethical approval by the Biology Animal Welfare and Ethical Review Board, University of York. Ethical approval for individual study sites is provided in electronic supplementary material, S1B.

Data accessibility. Excel file of data and R scripts used for analyses provided in the electronic supplementary material [67].

Authors' contributions. E.K.: Design of research questions, data collation and formatting, data analysis, writing manuscript drafts and leading editing process. K.S.: Design of research questions, intellectual advice and supervision, editing manuscript. A.L.: Design of research questions, and editing manuscript. S.E.S.: Statistical analysis and feedback on the manuscript. K.E.G.: Data coding and feedback on the manuscript. S.M.: Literature search on vocal repertoires and feedback on the manuscript. All other authors: data collection and feedback on the manuscript.

Competing interests. The authors declare no competing interests.

Funding. No funding was provided specifically for the current paper, but funding which supported data collection at field sites is acknowledged in electronic supplementary material, S11.

Acknowledgements. This highly collaborative project would not be possible without the many fieldwork teams and funding provided for data collection across all field sites. We name these and others in electronic supplementary material, S11, and we would also like to thank the Amboseli Baboon Project, Elizabeth Archie, Susan Alberts and Jake Gordan for contributing data on yellow baboons.

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
