## [Peer Review File · Royal Society Open Science]

Review History

Decision letter (RSOS-210873.R0)

Dear Ms Kavanagh

On behalf of the Editors, we are pleased to inform you that your Manuscript RSOS-210873 "Dominance style is a key predictor of vocal use and evolution across nonhuman primates" has been accepted for publication in Royal Society Open Science subject to minor revision in accordance with the referees' reports. Please find the referees' comments along with any feedback from the Editors below my signature.

Please submit your revised manuscript and required files (see below) no later than 7 days from today's (ie 09-Jun-2021) date. Note: the ScholarOne system will 'lock' if submission of the revision is attempted 7 or more days after the deadline. If you do not think you will be able to meet this deadline please contact the editorial office immediately.

Kind regards,
Royal Society Open Science Editorial Office
Royal Society Open Science
opencience@royalsociety.org

on behalf of Dr Alecia Carter (Associate Editor) and Kevin Padian (Subject Editor)
opencience@royalsociety.org

Associate Editor Comments to Author (Dr Alecia Carter):
Comments to the Author:

Dear authors,

Thank you for submitting your manuscript to RSOS. I have gone through the responses to the reviewers, and read the manuscript myself, and I am satisfied that the responses are considered and complete. I enjoyed reading the manuscript and can only add to the reviewers' compliments about generating such an impressive collaborative dataset and the interesting findings (both positive and null) that came out of the analyses.

On reading the manuscript, I found some minor grammatical points that could be incorporated in the manuscript (see below).

L119: a multiple levels of analysis study -> a multiple-levels-of-analysis study

L172: individual and species level -> individual- and species-level

L209: frequently used -> frequently-used

L210: why the mix of commas and semi-colons in the parentheses? Best to stick to just commas?

L265, 313: a list should start with a colon (not a semi-colon). It would be helpful to number the three measures as the sentence is complex and difficult to follow.

L270: by only including -> by including only

L275 known which -> known, which

Table 1: there should be a space between digits and units

L323: space after opening bracket

L324: what is "aggression intensity I"?

L325: group and species level random effects -> group and species as random effects (please clarify: were these nested?)

L331, 332: insert spaces between digits and symbols

Throughout: hyphenate compound nouns e.g. species level -> species-level (as is written at L341 cf. Table 2, etc)

L360: can you please provide some descriptive statistics (or a figure?) summarising the variation in vocal rates, etc.? (This information is depicted at the species level in Figure 2; readers will also be curious about within-species variation.)

L360: formatting?

Throughout: please be consistent using brackets for numbered lists: sometimes both opening and closing parentheses are used, others only the closing one. (FWIW, I prefer and find it easier to read when both are used.)

===PREPARING YOUR MANUSCRIPT===

Please also ensure that you include a Conflict of Interest statement as, upon submission, one of your authors is/was a *Royal Society Open Science* Editorial Board member. We suggest the following phrasing: "At the time of writing, [PROFESSOR NAME HERE] is a Board Member of Royal Society Open Science, but had no involvement in the review or assessment of the paper."

If you have been asked to revise the written English in your submission as a condition of publication, you must do so, and you are expected to provide evidence that you have received language editing support. The journal would prefer that you use a professional language editing service and provide a certificate of editing, but a signed letter from a colleague who is a native speaker of English is acceptable. Note the journal has arranged a number of discounts for authors

using professional language editing services
(<https://royalsociety.org/journals/authors/benefits/language-editing/>).

===PREPARING YOUR REVISION IN SCHOLARONE===

-- If you have uploaded ESM files, please ensure you follow the guidance at <https://royalsociety.org/journals/authors/author-guidelines/#supplementary-material> to include a suitable title and informative caption. An example of appropriate titling and captioning may be found at https://figshare.com/articles/Table_S2_from_ls_there_a_trade-

off_between_peak_performance_and_performance_breadth_across_temperatures_for_aerobic_sc
ope_in_teleost_fishes_/3843624.

Author's Response to Decision Letter for (RSOS-210873.R0)

See Appendix A.

Decision letter (RSOS-210873.R1)

Dear Ms Kavanagh,

I am pleased to inform you that your manuscript entitled "Dominance style is a key predictor of vocal use and evolution across nonhuman primates" is now accepted for publication in Royal Society Open Science.

You can expect to receive a proof of your article in the near future. Please contact the editorial office (openscience@royalsociety.org) and the production office (openscience_proofs@royalsociety.org) to let us know if you are likely to be away from e-mail contact – if you are going to be away, please nominate a co-author (if available) to manage the proofing process, and ensure they are copied into your email to the journal. Due to rapid publication and an extremely tight schedule, if comments are not received, your paper may experience a delay in publication.

Kind regards,
Royal Society Open Science Editorial Office
Royal Society Open Science

on behalf of Dr Alecia Carter (Associate Editor) and Kevin Padian (Subject Editor)
openscience@royalsociety.org

Appendix A

RESPONSE TO DECISION LETTER

Dear authors,

Thank you for submitting your manuscript to RSOS. I have gone through the responses to the reviewers, and read the manuscript myself, and I am satisfied that the responses are considered and complete. I enjoyed reading the manuscript and can only add to the reviewers' compliments about generating such an impressive collaborative dataset and the interesting findings (both positive and null) that came out of the analyses.

We very much appreciate these positive comments.

On reading the manuscript, I found some minor grammatical points that could be incorporated in the manuscript (see below).

L119: a multiple levels of analysis study -> a multiple-levels-of-analysis study

L172: individual and species level -> individual- and species-level

L209: frequently used -> frequently-used

These have all been corrected in the revised manuscript (line numbers unchanged)

L210: why the mix of commas and semi-colons in the parentheses? Best to stick to just commas?

These have been changed to commas (line numbers unchanged)

L265, 313: a list should start with a colon (not a semi-colon). It would be helpful to number the three measures as the sentence is complex and difficult to follow.

This paragraph has been re-structured to facilitate numbering the three types of measures in a clear way that is consistent with the numberings used throughout the manuscript.

L270: by only including -> by including only

This has been changed (L273 and L275)

L275 known which -> known, which

Changed; now L270

Table 1: there should be a space between digits and units

This has been corrected

L323: space after opening bracket

The space has been removed (L323)

L324: what is "aggression intensity I"?

This was a typo - "I" has been removed.

L325: group and species level random effects -> group and species as random effects (please clarify: were these nested?)

This has been changed, and clarified that they were nested (L325)

L331, 332: insert spaces between digits and symbols

This has been corrected

Throughout: hyphenate compound nouns e.g. species level -> species-level (as is written at L341 cf. Table 2, etc)

These have been changed throughout

L360: can you please provide some descriptive statistics (or a figure?) summarising the variation in vocal rates, etc.? (This information is depicted at the species level in Figure 2; readers will also be curious about within-species variation.)

We think this is a great idea, so we have now added this sentence to the start of the results (L360)

"Description of the variation within species in terms of vocal rate and dominance style measures can be found in tables S6-B in the supplementary material."

The new table added to the supplementary material provide interested readers with the mean, standard deviation and range of the vocal rate and each of the dominance style measures for each group within each species.

L360: formatting?

This formatting error has been corrected (L360)

Throughout: please be consistent using brackets for numbered lists: sometimes both opening and closing parentheses are used, others only the closing one. (FWIW, I prefer and find it easier to read when both are used.)

These have been changed throughout to include both brackets.